# Language Models Meet World Models: Embodied Experiences Enhance Language Models

**Jiannan Xiang**[*♠], **Tianhua Tao**[*♣], **Yi Gu**[♠], **Tianmin Shu**[◇△],
**Zirui Wang**[♠], **Zichao Yang**[♡], **Zhiting Hu**[♠]
[♠]UC San Diego, [♣]UIUC, [◇]MIT, [△]JHU, [♡]CMU

## Abstract

While large language models (LMs) have shown remarkable capabilities across numerous tasks, they often struggle with simple reasoning and planning in physical environments, such as understanding object permanence or planning household activities. The limitation arises from the fact that LMs are trained only on written text and miss essential embodied knowledge and skills. In this paper, we propose a new paradigm of enhancing LMs by finetuning them with *world models*, to gain diverse embodied knowledge while retaining their general language capabilities. Our approach deploys an embodied agent in a world model, particularly a simulator of the physical world (VirtualHome), and acquires a diverse set of embodied experiences through both goal-oriented planning and random exploration. These experiences are then used to finetune LMs to teach diverse abilities of reasoning and acting in the physical world, *e.g.*, planning and completing goals, object permanence and tracking, *etc.* Moreover, it is desirable to preserve the generality of LMs during finetuning, which facilitates generalizing the embodied knowledge across tasks rather than being tied to specific simulations. We thus further introduce the classical *elastic weight consolidation* (EWC) for selective weight updates, combined with *low-rank adapters* (LoRA) for training efficiency. Extensive experiments show our approach substantially improves base LMs on 18 downstream tasks by 64.28% on average. In particular, the small LMs (1.3B, 6B, and 13B) enhanced by our approach match or even outperform much larger LMs (e.g., ChatGPT). [1]

## 1 Introduction

Language Models (LMs) have demonstrated impressive performance on a wide range of natural language processing tasks [34, 48, 4, 7, 54]. In particular, recent studies show that LMs can assist decision-making for embodied tasks [1, 18, 25, 45, 19], demonstrating a certain level of understanding of the physical world. However, such understanding is not robust enough for many reasoning and planning tasks in physical environments. As shown in Figure 1, even the latest large LMs like ChatGPT[2] can still make mistakes in seemingly simple inquiries, such as counting objects in a location. We hypothesize that this is because current LMs trained merely with large-scale text corpora are devoid of embodied experiences such as navigating in an environment, interacting with objects, and sensing as well as tracking the world state. Consequently, they lack robust and comprehensive embodied knowledge necessary for reasoning and planning associated with physical environments. A related line of research finetunes LMs in order to improve specific embodied tasks, resulting in task-specialized models [6, 58, 21, 57].

---

[*]Equal contribution.

[1]The code is available at `https://github.com/szxiangjn/world-model-for-language-model`.

[2]Based on GPT-3.5-turbo.

37th Conference on Neural Information Processing Systems (NeurIPS 2023).

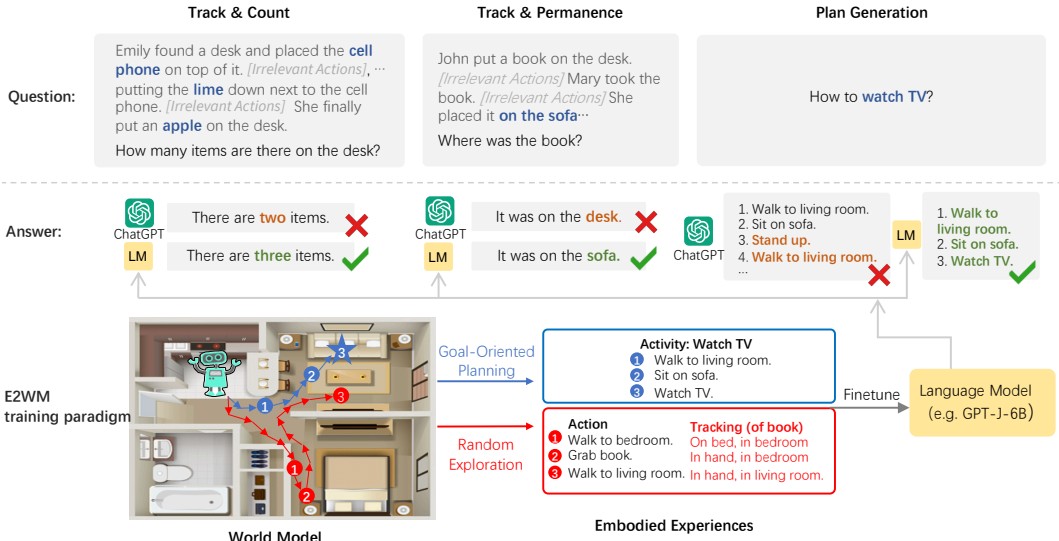

Figure 1: Examples of tasks requiring embodied knowledge (**upper**), and an overview of our approach (**bottom**). In the task examples, blue text indicates the useful information for answering the question.

In this paper, we aim to inject diverse fundamental embodied knowledge and skills into pretrained LMs, while retaining the models' generality. We introduce a novel training paradigm for LMs – fine-tuning with Embodied Experiences from World Models (E2WM). Here, world models are embodied simulators that emulate physical interactions in real-world environments (e.g., VirtualHome [36]). They provide LMs with the opportunity to comprehend object interactions within the environment and to execute actions, thus enabling a level of active engagement previously unattainable. These world models serve as a simplified and cost-effective replica of our real world that can significantly augment the conventional pretraining paradigm. We anticipate that finetuning LMs on embodied experiences gathered from world models can enhance their embodied knowledge—and, with the preserved model generality—consequently strengthen their abilities to solve a broad range of embodied tasks.

In this work, we consider a diverse range of fundamental knowledge and skills for embodied tasks, including tracking objects, planning to complete given goals, recognizing other agents' behaviors, *etc*. To this end, we introduce two ways to collect embodied experiences from world models that give rise to the desired knowledge and skills: **goal-oriented planning** and **random exploration** (Figure 1). Specifically, goal-oriented planning aims to gather experiences associated with planning and goal-oriented agent behaviors, while random exploration focuses on accumulating experiences that involve object and world state tracking. In goal-oriented planning, models are given the goal (*e.g.*, `IN(dust, trash can)`) for a specific activity (*e.g.*, `Clean Floor`), and is supposed to generate a plan to complete it. To find the plan, we devise Monte Carlo Tree Search (MCTS) [5, 44] to explore the world model. Then the process will be stored as an embodied experience. In random exploration, one or more agents are deployed in the world model to execute random actions, while the locations and the movements of all the objects are tracked simultaneously.

After collecting the embodied experiences, we use them to construct a set of fine-tuning tasks (e.g., plan generation, activity recognition, and tracking). Crucially, to finetune LMs on the collected embodied experiences while retaining their original general knowledge and capabilities, we propose to incorporate the classical Elastic Weight Consolidation (EWC) [22] into our training paradigm. By regularizing the finetuning loss, EWC aims to preserve the important LM parameters from pretraining. We show that EWC is substantially more effective than the popular KL regularization [35, 28, 31]. We further introduce efficient low-rank updates by harmonizing the recent Low-Rank Adaptation (LoRA) [16] with the EWC regularizer. This results in the new EWC-LoRA update rule that greatly reduces training costs and makes our E2WM paradigm accessible to cheap hardware (GPUs).

We instantiate a world model using a virtual household simulator, VirtualHome [36, 37], and apply our method to GPT-Neo-1.3B [3], GPT-J-6B [49], OPT-13B [59], and LLaMA-13B [48] models. To test the generalizability of the finetuned LMs, we evaluate them on a variety of unseen tasks which

demand similar embodied knowledge required to solve the training tasks. Additionally, we assess the models' performance on the original pretraining data to determine the extent to which their core language modeling abilities are retained. Experiments show that our method significantly improves the baselines on both seen and unseen tasks (*e.g.*, $34.31 \to 51.23$ Rouge-L on plan generation task, $30.41\% \to 67.01\%$ accuracy on counting task), without suffering performance degradation on the pretraining dataset ($3.443 \to 3.537$ perplexity on Pile test subset [12]). Moreover, the small GPT-J-6B, OPT-13B, and LLaMA-13B models finetuned with our E2WM paradigm even outperforms ChatGPT on many of the tasks. The experimental results demonstrate the effectiveness of E2WM as a promising fine-tuning mechanism to enhance pretrained LMs with generalizable embodied knowledge and skills.

## 2 Related Work

**World Model.** The term "world model" generally refer to a computational representation of the physical world, capable of simulating changes in the world's state in response to various actions. For instance, humans possess an internal world model that aids in predicting the outcomes of specific actions during the planning process. Recent research induces world models from large LMs for robust human-like reasoning [15]. In this work, we employ a simulator equipped with a physics engine to serve as our world model, effectively emulating real-world conditions. In the field of embodied AI, various world models are built to replicate the real world and serve as virtual test environments for assessing robotic agents before real-world deployment. For example, VirtualHome [36, 37] is a simulated 3D household environment implemented by Unity3D game engine. AI2-THOR [23] consists of near photo-realistic 3D indoor scenes and has richer object attributes and interaction types. Other indoor household World Models include VRKitchen [13], CHALET [56], MINOS [41], House3D [53], *etc.* Besides, MineCraft is a more challenging and open-ended world model, which has a large number of objectives and a large-scale task hierarchy [14, 27, 20]. In this paper, we use VirtualHome as our world model.

**Language Model Grounding.** A significant number of recent works focused on grounding language models to world models [1, 24, 38, 47, 55]. Some of them freeze LMs and leverage certain prompting strategies or specifically-designed modules. For example, Zero-Shot Planner [18] prompts LMs to generate activity plans and translate them into admissible actions. Mind's eye [29] prompts LMs to do simulations with physical engines to answer physical reasoning questions. SayCan [1] uses a learned affordance function to assist LMs in selecting valid actions. DEPS [50] prompts LMs to describe, explain and generate action plans, incorporated with a learned selector module to choose the most efficient path. There are also other works finetuning LMs towards better downstream task performance. For example, Li et al. [25] finetune LMs with supervised learning for interactive decision making, and Carta et al. [6] ground LMs with online reinforcement learning. Different from these works aiming to optimize LMs for specific tasks in the target environments, our work instead focuses on improving the language model itself by acquiring knowledge from world models.

**Language Model Regularization.** To facilitate the acquisition of new knowledge and skills without losing LMs' language modeling abilities , regularization is often introduced during finetuning. One popular method is adding KL penalty [35, 28, 46, 52, 33, 60], which leverages KL divergence between the output probability of the currently trained model and the original model to regularize the LM in an RL manner, *i.e.*, by computing policy gradients. For example, InstructGPT uses KL penalty to mitigate over-optimization of the reward model [35], and Liu et al. [28] add KL regularization for training a commonsense knowledge generator. In this work, we instead use *elastic weight consolidation* (EWC) for regularization. Our empirical results demonstrate that EWC is more effective than applying KL penalty for retaining language modeling abilities and generality of LMs.

## 3 Approach

In this work, we propose a new training paradigm, namely finetuning with Embodied Experiences from World Models (E2WM), to inject embodied knowledge into LMs without sacrificing its generality and language modeling abilities. The world model we use is **VirtualHome** [36, 37], a multi-agent simulator for househould activities. In VirtualHome, an executable action step can be simplified as the format of `[action] <arg>`, *e.g.*, `[Grab] <apple>` . The world state of VirtualHome consists of objects and their relations (*e.g.*, apple on table). Details about VirtualHome can be found in Appendix

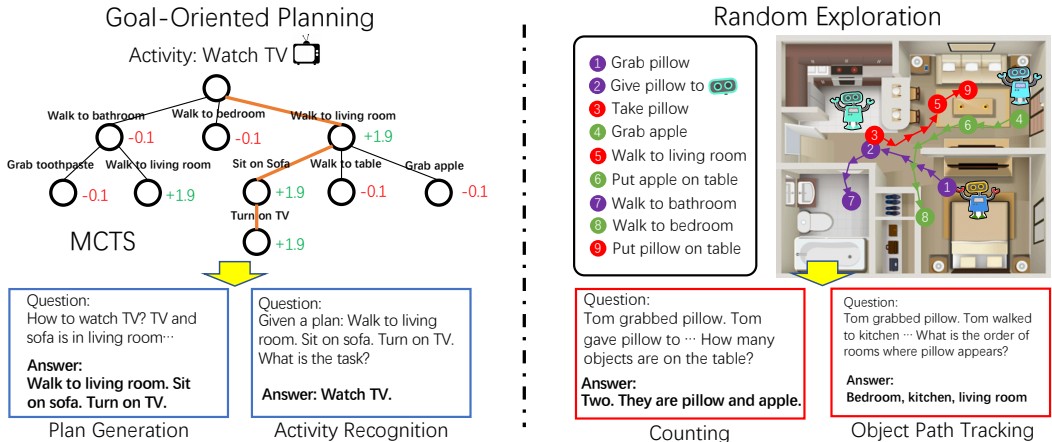

Figure 2: The illustration of goal-oriented planning (**left**) and random exploration (**right**) in our training paradigm. In MCTS, the path in orange represents the final plan generated by the planner.

A.1. We first describe how to gather embodied experiences in the world model in Section 3.1. Then in Section 3.2 we demonstrate how to finetune LMs by utilizing collected experiences, as well as our proposed method EWC-LoRA for efficient knowledge generalization.

## 3.1 Collecting Embodied Experiences from World Model

LMs pretrained on large scale human-written text corpus often have difficulties in solving basic reasoning and planning in physical environments. This is because that LMs lack necessary embodied knowledge and experiential understanding of the physical world. To address the problem, we propose to leverage world models to collect diverse embodied experiences for enhancing LMs. Specifically, to inject different types of embodied knowledge into LMs, we introduce two ways to gather experiences: goal-oriented planning and random exploration. Figure 2 illustrates the two methods.

**Goal-oriented Planning.** One important embodied skill is to plan and complete a specific goal, *e.g.*, placing tableware properly to set up the table. To endow LMs with this ability, we propose goal-oriented planning. The approach aims to generate experiences that are goal-oriented, thus are useful to facilitate the acquisition of skills and task planning abilities for executing a range of activities in the world model. To do that, we collect various activities and their corresponding goals. Formally, the goal for an activity in the world model is defined as a set of predicates describing the target world state. For instance, an activity can be `set up table`, and its goal can be `ON(fork, table);ON(plate, table)`, which means that fork and plate should be put on the table to fulfill the activity. More details about predicates and goal definitions can be found in Appendix A.2. As shown in Figure 2, in goal-oriented planning, we devise a Monte Carlo Tree Search (MCTS) planner to search through the action space and find a plan, *i.e.*, a sequence of actions, to achieve the goal. The key to successful MCTS is the reward design. At each time step, if at least one goal predicate is satisfied, the MCTS planner will get a reward of +2, and the achieved goal predicates will be removed from the goal. This ensures that the planner does not repeatedly execute the same action to receive rewards, but rather focuses on achieving the remaining unfulfilled goals. Besides, it will get a -0.1 penalty after each time step to discourage the planner from doing actions irrelevant to fulfilling the goals. Finally, we store the planning process as an embodied experience.

**Random Exploration.** In real-world scenarios, humans not only acquire new knowledge by finishing tasks, but also learn by just randomly exploring the surroundings, *e.g.*, randomly observing/tracking objects and knowing their properties. To mimic this learning process, we propose another approach, namely random exploration. By simply exploring in the world model, embodied experiences emerge that involve advanced cognitive abilities including object permanence and tracking, as agents observe and track the consistent existence of objects even when they are out of sight. Then the experiences are gathered for finetuning LMs later. Specifically, the approach deploys one or multiple agents in the world model wandering aimlessly and randomly executing actions. As illustrated in Figure 2, multiple agents are in the same environment, interacting with each other or executing different actions

on the same objects, which simulates complex situations. During the exploration, the moving paths and the final locations of all the objects in the world model are recorded. Then the whole process is captured as an embodied experience.

## 3.2 Finetuning LMs with Embodied Experiences

There are multiple ways to utilize collected embodied experiences for LMs finetuning, such as supervised learning and reinforcement learning. In this work, we use them with supervised learning for simplicity and efficiency. Specifically, goal-oriented planning experiences are compiled into data examples in two formats: **plan generation** and **activity recognition**. As shown in Figure 2, in plan generation, the model is required to generate a stepwise action sequence to fulfill an activity, given the state of some relevant objects as the initial condition. In activity recognition, the model needs to recognize the activity name given its plan. Experiences obtained from random exploration are also transformed into two self-supervision tasks: **counting** and **object path tracking**. Examples of the two tasks can be seen in Figure 2. Specifically, for counting, the LM is tasked with identifying the number and name of the objects at a specific location after the agents performed relevant and irrelevant actions and arranged objects randomly. In object path tracking, the model is tasked to output the moving path of an object that is picked up by different agents and moved to different rooms at different times. All the tasks are trained with cross-entropy loss. Suppose that $\mathbf{x}$ is the input (*e.g.*, the initial condition in plan generation) and $\mathbf{y} = \{y_1, ..., y_M\}$ is the label (*e.g.*, the stepwise action sequence), we finetune LMs by assigning different weights to different tasks:

$$\mathcal{L}_V = \sum_{v \in V} \alpha_v \sum_{m=1}^{M} \log P(y_m | \mathbf{y}_{<m}, \mathbf{x}), \tag{1}$$

where $\mathcal{L}$ is the loss function; $V$ is the task set; and $\alpha_v$ is the weight for task $v$. Following Flan-T5 [8], $\mathbf{x}$ is a prompt formatted to contain a task instruction and sampled in-context demonstrations. We provide all prompts in Appendix A.3.

**Efficient Finetuning with Preserved Generality.** However, there are two key problems for simply finetuning LMs. The first one is that LMs will easily overfit to the downstream tasks, leading to performance degradation on other tasks. This deviates from our goal that the model should generalize acquired knowledge across various tasks. Another problem is that finetuning the entire LM is resource-intensive and time-consuming, especially when the LM is extremely large. To overcome the problem and facilitate continual and efficient knowledge acquisition with world models, we propose to finetune only a small number of weights using low-rank adaptors (LoRA) [16] with elastic weight consolidation (EWC) [22], which we refer to as EWC-LoRA.

EWC is a regularization-based method typically used in the area of continual learning [9]. It calculates a fisher matrix [11] to estimate the importance of each parameter for a task and then uses it to regularize the training on a new task. The regularization term helps to constrain the parameter updates for the new task to avoid forgetting the previous knowledge. Let $U$ be the pretraining task set, and $V$ be the finetuning task set. Following [2], we have:

$$F_{i,i} = \frac{1}{N} \sum_{j=1}^{N} \left( \frac{\partial \mathcal{L}_U^{(j)}}{\partial \theta_{U,i}^*} \right)^2, \tag{2}$$

$$\mathcal{L}(\theta) = \mathcal{L}_V(\theta) + \lambda \sum_i F_{i,i} (\theta_i - \theta_{U,i}^*)^2, \tag{3}$$

where $\mathcal{L}$ is the loss function, $F$ is the fisher matrix, $\lambda$ is the hyperparameter, $i$ and $j$ are the indices for parameters and data samples, respectively, and $\theta$ and $\theta_U^*$ are currently trained parameters and frozen task $U$ parameters, respectively. Notice that the first term $\mathcal{L}_V(\theta)$ in Equation 3 is calculated in Equation 1, and the second term is the EWC regularizer. In Equation 2, the fisher matrix is calculated by averaging the sum of squares of the gradients from the task $U$, which indicates the significance of each parameter to the task $U$. Then the matrix is used in Equation 3 to weigh the shift of model parameters when training on $V$. By using EWC, the LM learns to adapt to new tasks without catastrophic forgetting on the pretraining task, which forces it to understand and digest new knowledge from the finetuning tasks instead of overfitting to them.

However, EWC is both time- and memory-inefficient. First, it requires finetuning the entire set of large LM's parameters. Moreover, the approach involves creating a frozen original model and a fisher matrix that is the same size as the LM, leading to a memory overhead of three times the original size.

This makes it particularly challenging to apply to larger models. To alleviate the problem, we propose to combine EWC with low-rank adaptors (LoRA), a parameter-efficient tuning method. LoRA freezes the pretrained model weights and injects two trainable low-rank matrices into each layer of the model. Suppose that $W, W^* \in \mathbb{R}^{r \times d}$ are the trained weight matrix and frozen weight matrix, respectively; and $B \in \mathbb{R}^{r \times k}, A \in \mathbb{R}^{k \times d}$ are two low-rank matrices with $k \ll \min(r, d)$. Then the formula for LoRA can be written as $W = W^* + BA$. Suppose that $H$ is flattened $BA$. Notably, we found that $\theta_i$ in Equation 3 is the element of $W$, and $\theta^*_{U,i}$ is that of $W^*$. Therefore, $\theta_i - \theta^*_{U,i}$ is the element of $H$. We can thus transform Equation 3 into the final formula of EWC-LoRA method:

$$\mathcal{L}(\theta) = \mathcal{L}_V(\theta) + \lambda \sum_i F_{i,i} h_i^2, \tag{4}$$

where $h_i = \theta_i - \theta^*_{U,i}$ is the $i$-th element of $H$. One of the benefits of this rewriting is that we no longer need to store the trained LM weight matrixes as what vanilla EWC does, which saves plenty of memory space. Besides, we only need to update $B$ and $A$ during the finetuning, which also lowers memory requirements and leads to much faster training speed. Surprisingly, as shown later, we empirically found that adding LoRA into EWC can further mitigate the issue of catastrophic forgetting and overfitting. This aligns with the previous conclusion that limiting the dimension of the optimization problem can alleviate catastrophic forgetting [32].

## 4 Experiments

**Training Details.** For goal-oriented planning, we collected activities and their corresponding target goals with data from RobotHow [36], a housework activity knowledge base created in VirtualHome. We applied our method to GPT-Neo-1.3B [3], GPT-J-6B [49], OPT-13B [59], and LLaMA-13B [48]. To save computing resources, we use Int8 technique [10]. Both of the models were trained with the AdamW optimizer [30]. All the hyperparameters are chosen according to the performance on a held out set. We used one NVIDIA GeForce RTX 3090 for training. More details can be found in Appendix A.4.

### 4.1 Downstream Evaluation Tasks

We developed various downstream evaluation tasks for each type of embodied knowledge, including both the training tasks as well as novel tasks unseen during training used for generalization evaluation. Additionally, we evaluate our models on bAbI [51], a dataset for testing multiple types of knowledge and abilities including embodied knowledge, logic reasoning, linguistic knowledge, *etc.* We select the bAbI tasks related to embodied knowledge for our evaluation. We evaluate all the unseen tasks including bAbI under few-shot settings, specifically 2-10 shots, by providing a few in-context exemplars in the prompts. We discuss more details of the tasks below.

**Plan Generation.** To evaluate planning ability, we construct downstream tasks using human-written plans from RobotHow. Specifically, we have three tasks:

- **Plan Generation Evaluation.** In this task, the model needs to generate a plan for a housework activity. It is similar to the training task but uses human-written plans as the ground truth instead of the collected experiences. We include activities unseen during training to test the generalizability of the model. Inspired by the previous study showing that LMs can easily be distracted by irrelevant context [43], we also created samples having states of unrelated objects in the context (*e.g.*, `TV is on` for activity `Make Coffee`) to confuse LMs. In summary, this results in four settings: Vanilla Seen, Vanilla Unseen, Confusing Seen, and Confusing Unseen. We have 175/54/135/43 examples for four settings, respectively. We use Rouge-L [26] as the metric.

- **Housework QA.** This is a multi-choice QA task, which is unseen during training. It asks which choice is the relevant object to finish a household activity, *e.g., which object is relevant to making coffee?* It has 261 examples in total, and we use accuracy as the metric. When evaluating, we provide 10 in-context exemplars in the prompts, so this task is evaluated as a 10-shot learning task.

- **Negation Housework QA.** This is similar to Housework QA but inquires about the irrelevant object, *e.g., which object is irrelevant to making coffee?* It is more challenging than the vanilla QA because LMs that simply memorize the word co-occurrence in the training data may succeed in the vanilla QA but will fail in the negation QA. This task has 162 examples and uses accuracy as the metric. We provide 10 in-context exemplars in the prompts.

**Activity Recognition.** We developed two multi-choice QA tasks with the human-written plans and the state changes from RobotHow to test the knowledge gained from activity recognition:

- **Activity Recognition QA.** In this task, a human-written plan is given and the model needs to choose the correct activity name. An example of the question is *"Given a plan: Walk to living room. Sit on sofa. Turn on TV. What is the name of this activity?"*. And the answer should be *Watch TV*. The task has 549 examples. We use accuracy as the metric.

- **Activity Inference QA.** In this task, we use the final state of the world model as input for the model to infer the activity name. For example, the input can be *"Tom is sitting on the sofa and facing the TV. The TV is on. What is a possible activity he is doing?"*, and the answer is *"Watch TV"*. We have 262 examples for this task and use accuracy to measure the performance. We provide 10 in-context exemplars in the prompts.

**Counting.** We gathered random exploration experiences to construct **Counting QA** for evaluating skills learned from the counting task. The model is required to answer the number of objects in a specific location. For example, a query can be *"Tom put an apple on the table. Tom turned on the TV. Tom put a cup on the table. How many objects are there on the table?"*. We can see that there will be irrelevant actions like *turn on TV* to confuse the model and make the question more challenging. We collected 194 samples for the task and used accuracy as the evaluation metric. We provide 5 in-context exemplars in the input prompts.

**Object Path Tracking.** We developed two downstream tasks for the object path tracking training task, namely Object Path Tracking Evaluation and Object Location QA.

- **Object Path Tacking Evaluation.** This evaluation task is the same as the training task, where the model is required to generate the full moving path of an object. An example is *"Tom walked to the kitchen. Tom grabbed the apple. Mary walked to the bedroom. Tom walked to the living room. What is the order of the rooms where the apple appears?"*. This question typically includes multiple agents and many irrelevant actions, which makes it difficult to track the object. This task contains 200 examples. Following Huang et al. [18], we evaluate the performance by calculating the length of the longest common subsequence (LCS) between the ground truth and the generated path, normalized by the maximum length of the two.

- **Object Location QA.** In this task, the model is asked about the location of an object before/after it moves to another location, *e.g., where is the apple before/after the kitchen?* This task has 200 examples with accuracy as the metric. We provide 2 in-context exemplars in the prompts.

A previous study on prompting multiple QA questions [39] introduces two prompting methods, multiple choice prompt and cloze prompt, and two normalization methods, length, and unconditioned normalization. For all the multi-choice QA tasks, we choose the combination of prompting and normalization methods which yields the best performance on a held out set.

To further verify the effectiveness of our method, we evaluate our finetuned GPT-Neo and GPT-J on the **bAbI** dataset. Specifically. we select 8 test sets from bAbI that align with the abilities covered in our collected embodied experiences. We include the description of each test set in Appendix A.5. For all the bAbI tasks, we do 2-shot learning by providing 2 in-context exemplars in the input prompts.

Besides downstream tasks, we also want to ensure that our approach does not hurt language modeling performance of the models. Therefore, following previous work [42], we evaluate the **perplexity** on a subset of Pile [12] test set, which is the pretraining dataset for GPT-Neo and GPT-J. We sampled 5000 examples from Pile test set for evaluation.

## 4.2 Results

**Constructed Evaluation Tasks.** Results for all the downstream evaluation tasks are shown in Figure 3 and Figure 4. For all the models, we compare the results obtained after finetuning with world model against those of the original base models. For GPT-J, we also include a finetuned model without EWC-LoRA as a baseline. Detailed numbers of the results can be found in Appendix A.6. We also conduct human evaluations for GPT-J on the plan generation task, which can be found in Appendix A.7. In general, the models trained with the world model significantly outperform the baselines on various downstream tasks. Our method is not only effective for small 1.3B model, but can also scale to larger 6B and 13B models. Specifically, our finetuned GPT-J and LLaMA-13B

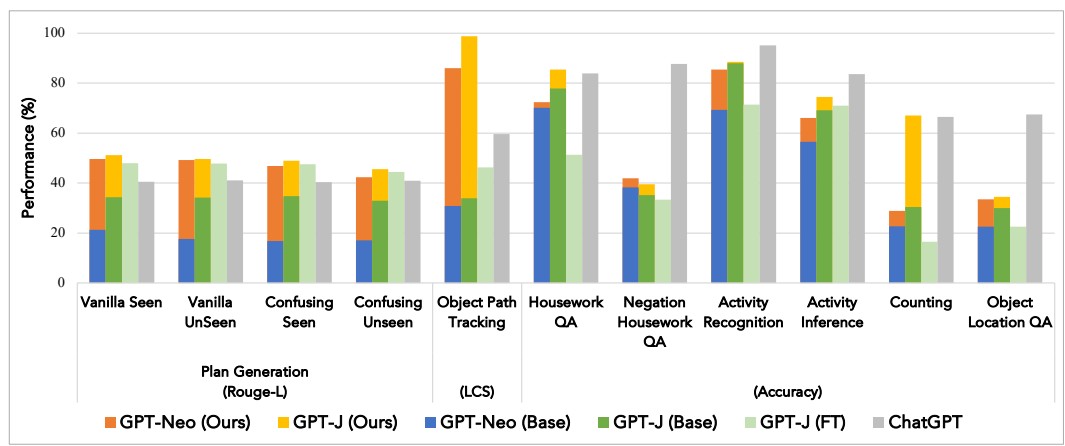

Figure 3: Experimental results of GPT-Neo and GPT-J on our constructed downstream tasks. GPT-J (FT) refers to the finetuned GPT-J without EWC-LoRA. Our approach surpasses baselines on all of the 11 tasks, and outperforms ChatGPT on 7 of them. For example, our GPT-J model achieves 98.67 LCS on object path tracking, which is significantly better than 33.86 of base GPT-J and 59.53 of ChatGPT.

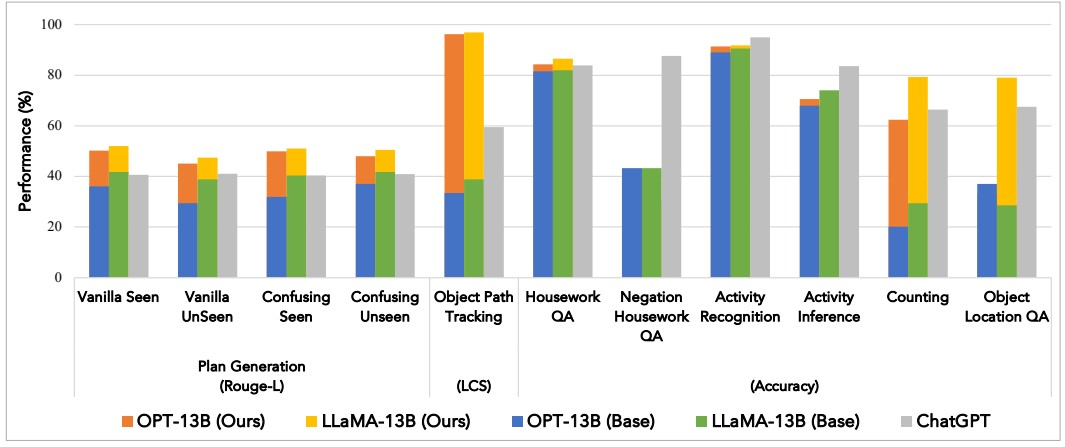

Figure 4: Experimental results of OPT-13B and LLaMA-13B on our constructed downstream tasks. Our approach applied on LLaMA-13B outperforms ChatGPT on 8 of them.

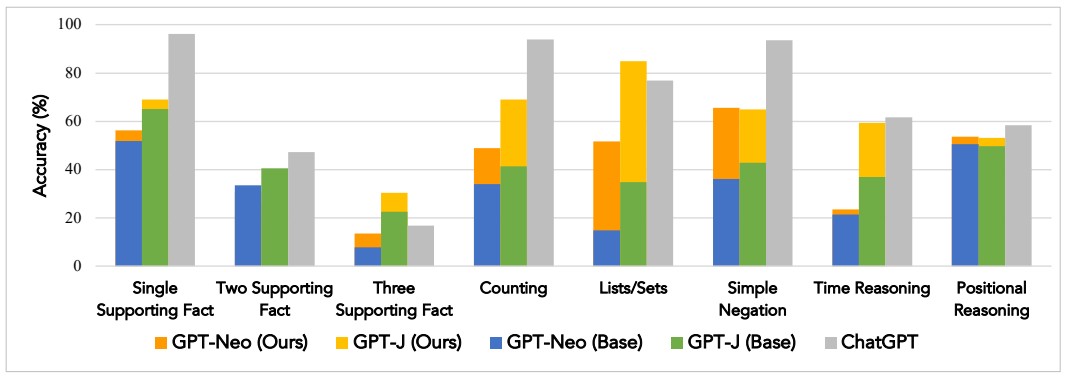

Figure 5: Experimental results on bAbI. Our approach outperforms base LMs on all the tasks except for the Two Supporting Fact task.

| | GPT-Neo | | GPT-J | | OPT-13B | | LLaMA-13B | |
|---|---|---|---|---|---|---|---|---|
| | Base | Ours | Base | Ours | Base | Ours | Base | Ours |
| | 4.120 | 4.193 | 3.443 | 3.537 | 4.077 | 4.358 | 3.036 | 3.069 |

Table 1: Perplexity on Pile test subset, showing the proposed finetuning with world model manages to preserve the LMs' language modeling capability.

| Task | GPT-Neo | | | | | GPT-J | | |
|---|---|---|---|---|---|---|---|---|
| | Base | EWC | LoRA | LoRA & KL | EWC-LoRA | Base | LoRA | EWC-LoRA |
| Plan Gen | 21.25 | 48.56 | **51.24** | 45.99 | 49.70 | 34.31 | **51.23** | 51.23 |
| Act Recog | 69.22 | **89.98** | 87.98 | 81.42 | 85.43 | 87.98 | **90.16** | 88.52 |
| Count | 22.68 | **55.67** | 27.84 | 49.48 | 28.87 | 30.41 | 63.92 | **67.01** |
| Obj PT | 30.80 | **95.96** | 87.28 | 63.59 | 85.91 | 33.86 | 97.22 | **98.67** |
| Perplexity | 4.120* | 4.995 | 4.360 | 5.029 | **4.193** | 3.443* | 3.675 | **3.537** |

Table 2: Results of different regularization methods. The abbreviations in the Task column stand for the corresponding evaluation tasks for four training tasks. We use asterisk* to mark the perplexity of base models.

with world model even achieve better performance than ChatGPT as a much larger LM on most of the 11 tasks. Besides, we can see the world model improves LMs on both seen and unseen tasks. This demonstrates that our model absorbs the knowledge for goal-oriented planning and random exploration instead of memorizing the seen experiences. Specifically, the better plan generation performance under the "Confusing" setting indicates that the world model improves the ability of LMs to avoid being interfered with by irrelevant contexts. On both Housework QA and Negation Housework QA, our models surpass the baselines, showing that our models also acquire knowledge about the necessary objects for completing a housework activity. Results on other downstream tasks also prove the effectiveness of our method. For example, on both Activity Recognition Evaluation and Activity Inference, our approach improves over the baselines significantly. Moreover, improvements can be observed in the downstream tasks regarding random exploration. On Counting and Object Location QA tasks, our LLaMA-13B trained with the world model even surpasses ChatGPT.

**bAbI Tasks.** To further verify the effectiveness of our method, we evaluate our finetuned models on the bAbI dataset. The results are shown in Figure 5. We can see that our approach significantly outperforms the base LMs. Notably, after finetuned with VirtualHome experiences, GPT-J surpasses the much stronger ChatGPT on the most challenging tasks. Specifically, it outperforms ChatGPT on the Three Supporting Fact task, where the model is required to use three supporting facts from the context to answer a question like "*where was the apple before the kitchen?*", and Lists/Sets task, which asks the model to give the answers in the form of a list, *e.g.*, the answer for "*What is Daniel holding?*" is "*apple, milk*". These results prove that our approach enables LMs to acquire the knowledge and skills inherent in embodied experiences, rather than simply overfitting to the training environment.

**Language Modeling.** In addition to verifying improved performance on the downstream tasks, we also report results on Pile test subset to ensure the preservation of the generality and language modeling abilities of LMs. From the experimental results shown in Table 1, we can see that our approach only causes a negligible increase in perplexity over the base models. This demonstrates the effectiveness of EWC-LoRA to preserve the generality and linguistic competence of LMs. To verify the generality on other NLP tasks, we also include the results on SuperGLUE [40] in Appendix A.8.

### 4.3 Comparison of Different Regularization Methods

We compare our proposed EWC-LoRA with EWC and LoRA. Besides, we also include the baseline using KL penalty as regularization.The experimental results are shown in Table 2. We also include the results of four evaluation tasks. Notice that we do not include the results of GPT-J with pure EWC and KL, since they are overly memory-intensive or time-consuming. EWC requires an original model and a fisher matrix other than the trained model, which triples the memory usage, making

|          | Base   | Ours  | -w/o Plan Gen | GPT-Neo -w/o Act Recog | -w/o Count | -w/o Obj PT |
|----------|--------|-------|---------------|------------------------|------------|-------------|
| Plan Gen | 21.25  | 49.70 | 14.48         | 49.38                  | 49.85      | **50.06**   |
| Act Recog| 69.22  | 85.43 | **85.97**     | 48.63                  | 85.25      | 84.34       |
| Count    | 22.68  | 28.87 | 18.56         | 25.26                  | **35.05**  | 32.99       |
| Obj PT   | 30.80  | 85.91 | **92.13**     | 84.17                  | 86.46      | 29.90       |
| Perplexity | 4.120* | 4.193 | 4.171       | **4.151**              | 4.162      | 4.164       |

Table 3: Ablation experimental results on training tasks. We use the same abbreviations as Table 2.

it hard to be applied to large models like GPT-J-6B. Besides, KL penalty term is computed by $\mathcal{L}_{KL} = E_{(\mathbf{x},\mathbf{y}) \sim P_{\theta*}}[-\log\left(P_{\theta*}(\mathbf{y}|\mathbf{x})/P_{\theta}(\mathbf{y}|\mathbf{x})\right)]$, thus it requires sampling from the model output probability, which is time-consuming. On the contrary, EWC-LoRA is both memory- and time-efficient. In Table 2, we can see that EWC-LoRA achieves the lowest perplexity compared to other methods, while still significantly outperforming the base LMs. Compared with pure EWC, applying pure LoRA greatly decreases perplexity, which is consistent with the previous conclusion that limiting the dimension of the optimization problem can mitigate catastrophic forgetting [32]. EWC-LoRA further decreases perplexity, making it extremely close to the original perplexity, while achieving comparable performance with LoRA on downstream tasks. This demonstrates the effectiveness of EWC-LoRA. Besides, We can find that combing LoRA with KL will greatly increase perplexity while not achieving better downstream performance. Overall, our proposed EWC-LoRA achieves the best trade-off between the perplexity and the downstream performance, which outperforms baselines significantly while almost not increasing the perplexity on the pretraining dataset.

## 4.4 Ablation Studies

To study the contribution of each training task, we conducted an ablation study by removing one training task every time. We use GPT-Neo-1.3B as the base model. We include the results on tasks seen during training in Table 3. Results on all the tasks can be found in Appendix A. We can see that the removal of a training task with similar ability leads to a notable decrease in the model's performance on downstream tasks. For example, the performance of plan generation drops significantly when plan generation is removed from the training tasks. Similarly, the removal of activity recognition or object path tracking from the training tasks leads to a decline in performance in their respective downstream tasks. We conclude that our gathered embodied experience has a tremendous contribution to teaching the corresponding reasoning ability by finetuning. Interestingly, Counting QA performance shows an increase when counting is omitted from the training tasks, possibly because the ability of counting can be inferred from other training tasks.

## 5 Conclusion & Future Work

We proposed a new training framework that uses world models to enhance language models. It first collects embodied experiences from world models through both goal-oriented planning and random exploration. The experiences are then compiled into appropriate formats for LMs finetuning. We further introduce EWC-LoRA, which not only facilitates parameter-efficient tuning but also alleviates catastrophic forgetting and enables knowledge generalization. We show the strong performance of our method on a large number of downstream evaluation tasks.

This work demonstrates the advantage of panoramic learning with all forms of experience [17]. On the other hand, the present work is limited to a single household environment as the world model. In the future, we intend to study how to integrate embodied experiences from different world models and generalize knowledge learned from each world model to different domains.

**Acknowledgements.** This project is partially supported by DARPA ECOLE HR00112390063.

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

| Action | Template |
|---|---|
| [Find] <Object> | Find Object |
| [Walk] <Object> | Walk to Object |
| [Run] <Object> | Run to Object |
| [Sit] <Object> | Sit on Object |
| [StandUp] | Stand up |
| [Grab] <Object> | Grab Object |
| [Open] <Object> | Open Object |
| [Close] <Object> | Close Object |
| [Put] <Object_1> <Object_2> | Put Object_1 on Object_2 |
| [PutIn] <Object_1> <Object_2> | Put Ojbect_1 in Object_2 |
| [SwitchOn] <Object> | Switch/Turn on Object |
| [SwitchOff] <Object> | Switch/Turn off Ojbect |
| [Drink] <Object> | Drink Object |
| [TurnTo] <Object> | Turn to Object |
| [LookAt] <Object> | Look at Object |
| [Wipe] <Object> | Wipe Object |
| [PutOn] <Object> | Put on Object |
| [PutOff] <Object> | Put off Object |
| [Greet] <Object> | Greet Object |
| [Drop] <Object> | Drop Object |
| [Touch] <Object> | Touch Object |
| [Lie] <Object> | Lie on Object |
| [Pour] <Object_1> <Object_2> | Pour Object_1 into Object_2 |
| [Type] <Object> | Type Object |
| [Watch] <Object> | Watch Object |
| [Move] <Object> | Move Object |
| [Wash] <Object> | Wash Object |
| [Rinse] <Object> | Rinse Object |
| [Scrub] <Object> | Scrub Object |
| [Squeeze] <Object> | Squeeze Object |
| [PlugIn] <Object> | Plug in Object |
| [PlugOut] <Object> | Plug out Object |
| [Cut] <Object> | Cut Object |
| [Eat] <Object> | Eat Object |
| [Sleep] | Sleep |
| [WakeUp] | Wake up |

Table 4: Supported actions in VirtualHome and their corresponding text templates.

## A  Appendix

### A.1  VirtualHome

The complete format of an executable action step in VirtualHome is <char{char_id}> [Action] <Object> (Object_id). Specifically, char_id specifies which agent to execute the action when multiple agents are in the world model at the same time. Action should be a supported atomic action in VirtualHome. Object is the object with which the agent interacts. Each object in the environment is assigned an Object_id to distinguish it from others of the same object class. We designed a template for each action to transform them into natural text for LMs finetuing. The full list of executable actions can be found in Table 4. Note that in the list, we omit <char{char_id}> and (Object_id) for simplicity.

### A.2  Acitivity Goal And Predicate

The goal of an household activity in VirtualHome consists of several predicates. Each predicate represents a condition of one object or a relation between two objects. For example, OPEN(coffe

`maker)` means the coffee maker is open, and `ON(apple, table)` means an apple is on the table. The goal is only achieved when all the predicates are achieved. We collected activities and goals from RobotHow.

## A.3  Data Format and Prompts

Following Chung et al. [8], we use instructions with in-context exemplars as prompts. Specifically, the instruction, the question context, and the answer will be provided in each exemplar, and the full prompt will contain multiple such exemplars for in-context learning. The format of the data and the exemplar for each task is provided below.

### A.3.1  Plan Generation

**Data Example**

| Key | Value |
| --- | --- |
| activity | `watch TV` |
| condition | `living room, sofa, TV. The sofa and TV are in the living room.` |
| plan | `Walk to living room. Sit on sofa. Watch TV.` |

**In-context Exemplar**

```
Q: How to {{ activity }}? Given items include {{ condition }}
A: {{ plan }}
```

### A.3.2  Housework QA

**Data Example**

| Key | Value |
| --- | --- |
| activity | `watch TV` |
| choices | `[TV, coffee, bed, toothbrush]` |
| answer | `TV` |

**In-context Exemplar**

```
Question: To {{ activity }}, a possibly related item could be
Answer: {{ answer }}
```

### A.3.3  Negation Housework QA

**Data Example**

| Key | Value |
| --- | --- |
| activity | `watch TV` |
| choices | `[TV, sofa, living room, toothbrush]` |
| answer | `toothbrush` |

**In-context Exemplar**

```
Question: To {{ activity }}, an unrelated item could be
Answer: {{ answer }}
```

### A.3.4 Activity Recognition

**Data Example**

| Key | Value |
|---|---|
| plan | Walk to living room. Sit on sofa. Watch TV. |
| choices | [watch TV, make coffee, sleep, brush teeth] |
| activity | watch TV |

**In-context Exemplar**

```
Given a task plan: {{ plan }}
Question: what is the name of this task?
Answer: {{ answer }}
```

### A.3.5 Activity Inference

**Data Example**

| Key | Value |
|---|---|
| state | Tom is sitting on the sofa. Tom is facing the TV. |
| choices | [watch TV, make coffee, sleep, brush teeth] |
| activity | watch TV |

**In-context Exemplar**

```
{{ state }}
Question: given the above state, a possible activity could be
Answer: {{ answer }}
```

### A.3.6 Counting

**Data Example**

| Key | Value |
|---|---|
| movement | Tom was at home. He grabbed an apple and put it on the bookshelf. He then walked to the kitchen and srcub a plate. He went back to bookshelf and put the plate on it. |
| location | bookshelf |
| number | 2 |
| items | apple, plate |

**In-context Exemplar**

```
Given a sequence of actions in a house, and a question about what items are located
in a specific place. Answer the number of items and list the items.

Q: {{ movement }} How many items are there on the {{ location }}?
A: Ther are {{ number }} itmes on the {{ location }}. They are {{ items }}
```

### A.3.7 Counting QA

**Data Example**

| Key | Value |
| --- | --- |
| movement | Tom was at home. He grabbed an apple and put it on the bookshelf. He then walked to the kitchen and srcub a plate. He went back to bookshelf and put the plate on it. |
| location | bookshelf |
| number | 2 |

**In-context Exemplar**

```
Q: {{ movement }} How many items are there on the {{ location }}?
A: {{ number }}
```

### A.3.8 Object Path Tracking

**Data Example**

| Key | Value |
| --- | --- |
| movement | Tom went to the kitchen. Mary walked into the dining room. Tom grabbed a plate. Tom travelled to the living room. Mary moved to the living room. Tom put the plate on the table. Mary grabbed the plate. Mary journeyed to the bedroom. |
| object | plate |
| path | kitchen, living room, bedroom |

**In-context Exemplar**

```
{{ movement }}
Question: What is the order of the rooms where the {{ object }} appeared?
Answer: {{ path }}
```

### A.3.9 Object Location QA

**Data Example**

| Key | Value |
| --- | --- |
| movement | Tom went to the kitchen. Mary walked into the dining room. Tom grabbed a plate. Tom travelled to the living room. Mary moved to the living room. Tom put the plate on the table. Mary grabbed the plate. Mary journeyed to the bedroom. |
| object | plate |
| reference_room | living room |
| preposition | before |
| answer | kitchen |

**In-context Exemplar**

```
{{ movement }}
Question: Where is the {{ object }} {{ preposition }} the {{ reference_room }}?
Answer: {{ answer }}
```

## A.4 Hyperparameters

For both GPT-Neo-1.3B and GPT-J-6B, we use a learning rate of $8 \times 10^{-5}$ and a batch size of 20. The weights for plan generation, activity recognition, counting, and object path tracking are 1.0, 0.7, 1.0, and 1.0, respectively. We trained GPT-Neo-1.3B for 3 epochs with the EWC coefficient $\lambda = 0.5$ in Equation 4. For GPT-J-6B, we trained it for 5 epochs with $\lambda = 2$. With our approach, it takes 40 minutes to train a GPT-Neo and 220 minutes to train a GPT-J. We used a rank of 8 and coefficient of 32 for LoRA's hyperparameters.

## A.5 bAbI Dataset

We include 8 tasks from bAbI that test embodied knowledge. They are: One Supporting Fact, Two Supporting Fact, Three Supporting Fact, Counting, Lists/Sets, Simple Negation, Time Reasoning, Positional Reasoning. Examples for each task are shown in Table 5.

| **Task 1: Single Supporting Fact** | **Task 2: Two Supporting Facts** |
|---|---|
| Mary went to the bathroom. 
 John moved to the hallway. 
 Mary travelled to the office. 
 Where is Mary? A:office | John is in the playground. 
 John picked up the football. 
 Bob went to the kitchen. 
 Where is the football? A:playground |

| **Task 3: Three Supporting Facts** | **Task 4: Counting** |
|---|---|
| John picked up the apple. 
 John went to the office. 
 John went to the kitchen. 
 John dropped the apple. 
 Where was the apple before the kitchen? A:office | Daniel picked up the football. 
 Daniel dropped the football. 
 Daniel got the milk. 
 Daniel took the apple. A: office 
 How many objects is Daniel holding? A: two |

| **Task 5: Lists/Sets** | **Task 6: Simple Negation** |
|---|---|
| Daniel picks up the football. 
 Daniel drops the newspaper. 
 Daniel picks up the milk. 
 What is Daniel holding? milk, football | Sandra travelled to the office. 
 Fred is no longer in the office. 
 Is Fred in the office? A:no 
 Is Sandra in the office? A:yes |

| **Task 7: Time Reasoning** | **Task 8: Positional Reasoning** |
|---|---|
| In the afternoon Julie went to the park. 
 Yesterday Julie was at school. 
 Julie went to the cinema this evening. 
 Where did Julie go after the park? A:cinema 
 Where was Julie before the park? A:school | The triangle is to the right of the blue square. 
 The red square is on top of the blue square. 
 The red sphere is to the right of the blue square. 
 Is the red sphere to the right of the blue square? A:yes 
 Is the red square to the left of the triangle? A:yes |

Table 5: Examples for bAbI tasks.

## A.6 Results of Main Experiments and Ablation Studies

Experimental results on our constructed downstream tasks are shown in Table 6, and the results on bAbI are shown in Table 7. We also show the results of ablation studies in Table 8.

## A.7 Human Evaluations

We conduct human evaluations on plan generation for GPT-J model. Following Huang et al. [18] we asked 3 people to annotate whether each task can be completed using a generated plan. We randomly sampled 150 tasks and asked each person to annotate 50 of them. The Results show that the base GPT-J model can only achieve 24.0% accuracy, while the finetuned model can achieve 62.4%. The higher planning accuracy demonstrates the superior task planning ability of our model.

## A.8 SuperGLUE Results

We evaluate the base GPT-J model and our finetuned model on appropriate SuperGLUE tasks, *e.g.*, that can be formulated as a multi-choice QA task without prompt engineering. Our model's performance matches and even outperforms the baseline, showing our model retains the general language capability.

| Task | Metric | GPT-Neo | | GPT-J | | | OPT-13B | | LLaMA-13B | | ChatGPT |
|---|---|---|---|---|---|---|---|---|---|---|---|
| | | Base | Ours | Base | FT | Ours | Base | Ours | Base | Ours | *(GPT3.5-turbo)* |
| Plan Generation | | | | | | | | | | | |
|   *-Vanilla Seen* | Rouge-L | 21.25 | 49.70 | 34.31 | 47.98 | 51.23 | 36.00 | 50.15 | 41.77 | **52.05** | 40.57 |
|   *-Vanilla UnSeen* | Rouge-L | 17.64 | 49.27 | 34.22 | 47.86 | **49.58** | 29.34 | 45.11 | 38.78 | 47.44 | 41.01 |
|   *-Confusing Seen* | Rouge-L | 16.86 | 46.88 | 34.81 | 47.59 | 48.94 | 31.92 | 49.87 | 40.33 | **51.00** | 40.41 |
|   *-Confusing Unseen* | Rouge-L | 17.05 | 42.34 | 32.98 | 44.43 | 45.60 | 36.98 | 47.93 | 41.73 | **50.49** | 40.97 |
| Housework QA | Accuracy | 70.11 | 72.41 | 77.78 | 51.34 | 85.44 | 81.61 | 84.29 | 81.99 | **86.59** | 83.91 |
| Negation Housework QA | Accuracy | 38.27 | 41.98 | 35.19 | 33.33 | 39.51 | **43.21** | 40.21 | **43.21** | 30.25 | 87.65 |
| Activity Recognition | Accuracy | 69.22 | 85.43 | 87.98 | 71.41 | 88.52 | 89.07 | 91.44 | 90.53 | **91.80** | 95.05 |
| Activity Inference | Accuracy | 56.49 | 66.03 | 69.08 | 70.99 | **74.43** | 67.94 | 70.61 | 74.05 | 68.32 | 83.59 |
| Counting | Accuracy | 22.68 | 28.87 | 30.41 | 16.49 | 67.01 | 20.01 | 62.37 | 29.38 | **79.38** | 66.49 |
| Object Path Tracking | LCS | 30.80 | 85.91 | 33.86 | 46.25 | **98.67** | 33.49 | 96.28 | 38.82 | 96.99 | 59.53 |
| Object Location QA | Accuracy | 22.50 | 33.50 | 30.00 | 22.50 | 34.50 | 37.00 | 33.00 | 28.50 | **79.00** | 67.50 |

Table 6: Experimental results on various downstream evaluation tasks. The best result among baselines and our method is shown in **bold**, and the best result among all the models is underlined.

| Task | GPT-Neo | | GPT-J | | ChatGPT |
|---|---|---|---|---|---|
| | Base | Ours | Base | Ours | |
| Single Supporting Fact | 51.86 | 56.29 | 65.16 | **68.98** | 96.27 |
| Two Supporting Fact | 33.43 | 30.82 | **40.48** | 26.08 | 47.33 |
| Three Supporting Fact | 7.85 | 13.49 | 22.46 | **30.41** | 16.82 |
| Counting | 34.04 | 48.84 | 41.39 | **69.08** | 93.96 |
| Lists/Sets | 14.80 | 51.76 | 34.74 | **84.99** | 76.84 |
| Simple Negation | 36.05 | **65.56** | 42.80 | 63.95 | 93.66 |
| Time Reasoning | 21.45 | 23.46 | 36.96 | **59.42** | 61.63 |
| Positional Reasoning | 50.51 | **53.64** | 49.70 | 53.23 | 58.38 |

Table 7: Experimental results on bAbI test sets.

| | | | | GPT-Neo | | |
|---|---|---|---|---|---|---|
| | Base | Ours | -w/o Plan Gen | -w/o Act Recog | -w/o Count | -w/o Obj PT |
| Plan Gen | | | | | | |
|   *-Vanilla / Seen* | 21.25 | 49.70 | 14.48 | 49.38 | 49.85 | **50.06** |
|   *-Vanilla / Unseen* | 17.64 | 49.27 | 14.28 | 48.96 | **51.16** | 49.02 |
|   *-Confusing / Seen* | 16.86 | 46.88 | 13.63 | 46.37 | 48.30 | **49.14** |
|   *-Confusing / Unseen* | 17.05 | 42.34 | 9.86 | 43.79 | **46.28** | 44.64 |
| QA | 70.11 | 72.41 | 73.18 | 71.26 | **74.71** | 70.11 |
| Neg QA | 38.27 | **41.98** | 32.72 | 35.80 | 36.42 | 38.89 |
| Act Recog | 69.22 | 85.43 | **85.97** | 48.63 | 85.25 | 84.34 |
| Act Infer | 56.49 | 66.03 | **66.03** | 58.40 | 64.89 | 62.21 |
| Count | 22.68 | 28.87 | 18.56 | 25.26 | **35.05** | 32.99 |
| Obj PT | 30.80 | 85.91 | **92.13** | 84.17 | 86.46 | 29.90 |
| Obj QA | 22.50 | 33.50 | 35.00 | **49.00** | 43.50 | 22.00 |
| Perplexity | 4.120* | 4.193 | 4.171 | **4.151** | 4.162 | 4.164 |

Table 8: Ablation experimental results on training tasks.

| Model | BoolQ | CB | RTE | AX-g | AX-b | COPA |
|---|---|---|---|---|---|---|
| GPT-J | | | | | | |
|   *- Base* | 45.20 | **41.07** | 47.29 | 50.00 | **57.50** | 59.00 |
|   *- Ours* | **66.00** | **41.07** | **58.84** | **53.37** | 54.00 | **62.00** |

Table 9: Results on SuperGLUE subset.

## A.9 Broader Impact

Like other generation systems, the language model trained by our approach is susceptible to producing unintended output when confronted with harmful input, such as unethical text or input intended for adversarial attacks. Therefore, we strongly advise against utilizing our approach outside of controlled research environments until these risks have been mitigated. It is important to note that a thoughtless deployment of our method could potentially enable malicious exploitation of the underlying language models. Thus, precautions, such as implementing a filtering mechanism, must be taken.

