# OpenReview forum: "Language Models Meet World Models: Embodied Experiences Enhance Language Models"
_NeurIPS.cc/2023/Conference — NeurIPS 2023 poster_

### Official Review · Reviewer_311c · 2023-06-30

**Soundness:** 3 good
**Presentation:** 3 good
**Contribution:** 3 good
**Rating:** 7
**Confidence:** 4

**Summary:**

This paper proposes a new collection of data and tasks from an embodied environment to enhance the embodied reasoning ability of pre-trained language models. The paper also designs a fine-tuning strategy that combines the advantages of EWC and LoRA for stable fine-tuning. The experiments show that the proposed method can efficiently fine-tune GPT-J and outperform much larger language models.

**Strengths:**

This paper designs two ways of data collection in the embodied environment that address goal-specific and general cases in real-world applications.

This paper designs a comprehensive reasoning task set based on the collected data to evaluate the reasoning performance from different aspects (plan generation, object tracking, etc.)

This paper adopts a suitable parameter updating strategy (EWC_LoRA) for language model fine-tuning, which demonstrates time and memory efficiency.

**Weaknesses:**

The paper lacks experiments on why EWC is needed. The motivation for including EWC is to avoid overfitting downstream tasks or catastrophic forgetting on the pretraining task, but LoRA has a similar purpose. From the experiments in Table 2, EWC_LoRA has a lower performance than LoRA except for a slight improvement in perplexity.

**Questions:**

1. Related to weakness: It will be good to see whether there are any other benefits gained from EWC.

2. How do you ensure that the answers of the negation Housework QA are truly irrelevant?

3. The activity recognition performance of all methods is quite similar. Does this indicate that the activity types are easily distinguishable without requiring a full description of the experience, for example, just keywords at the end are sufficient for the task? Have you tried to include confusing activities in the evaluation set?

4. How do you introduce the confusion term (counting task, confusing unseen of plan generation)? Do you have any guideline or do you just randomly inject phrases?

**Limitations:**

The author mentioned the limitations of only collecting embodied experience from one world model.

societal impact: did not find negative impacts, probably if the data collection is extended to the real world in the future, it should avoid collecting personal or sensitive experiences.

---

> ### Author Rebuttal · Authors · 2023-08-10
>
> We appreciate your constructive feedback. We would like to respond to your questions as below.
>
> **(1) Why EWC is needed**
>
> We want to clarify that EWC-LoRA has similar performance with LoRA. From Table 2, we can see EWC-LoRA matches or even outperforms its LoRA counterpart for GPT-J (e.g., 3.09 improvements on Counting QA, 1.45 on Object Path Tracking) and GPT-Neo (e.g., 1.03 on Count), while EWC-LoRA has a lower perplexity. Both methods outperform the base language model and have close performances on the specific tasks, in this case we focus more on the perplexity since we want its language modeling capability and generality to be preserved as much as possible.
>
> **(2) Irrelevant items in Negation QA**
>
> We use simple heuristics to sample irrelevant items, e.g., we sample items in the environment that are not directly mentioned in the ground truth plan to form the questions.
>
> **(3) Performance on Activity Recognition**
>
> For now, we just randomly sample different activities. The performances on this task are relatively higher than other tasks, but we can still observe significant performance gap of different models (69.22 by base GPT-Neo, compared to 87.98 by base GPT-J; see Table 6 in the appendix), and our method outperforms the base language model for both GPT-Neo (69.22 → 85.43) and GPT-J (87.98 → 88.52).
>
> **(4) Confusing terms**
>
> For the counting task, we construct samples from collected Random Exploration experiences, where an agent randomly executes actions in the environment. As a result, some of the actions are naturally irrelevant to object counting. For confusing unseen plan generation, we adopted the similar heuristics in (2).

---

> > ### Comment · Reviewer_311c · 2023-08-15
> >
> > Thank you for the explanation and the results of your model scale-up experiments in the rebuttal. I would like to maintain my rating. I believe a fine design of the evaluation tasks is more important than carefully tuned models. Therefore, I still recommend this work to the community.
> >
> > Additionally, I have the following comments:
> > 1. For Negation QA, if the irrelevant items are just a sample of things not mentioned in the ground truth plan, it seems that they cannot avoid being generally relevant. For example, "spoon" in the example of "which object is irrelevant to making coffee?". Since your evaluation set is relatively small (2 questions will affect 1% accuracy), it is better to have fewer ambiguous answers. As a benchmark task set, I would expect to see a more quantitative and systematic design, for instance, using n-hop relationships in the knowledge graph to construct the QA pairs.
> >
> > 2. Regarding the necessity of EWC, I have some doubts about it. It might be good to see the mean and standard deviation of multiple-round experiments.

---

> > > ### Author Response · Authors · 2023-08-18
> > >
> > > Thanks for your positive feedback and valuable insights! Your suggestions regarding the quantitative and systematic design of the benchmark set, as well as the multiple-round experiments for verifying the necessity of EWC, will greatly strengthen our work and improve our manuscript's presentation. We will include them in the revision.

---

### Official Review · Reviewer_7bDY · 2023-07-01

**Soundness:** 2 fair
**Presentation:** 2 fair
**Contribution:** 2 fair
**Rating:** 5
**Confidence:** 4

**Summary:**

The paper introduces a training paradigm termed "finetuning with Embodied Experiences from World Models (E2WM)" to enhance the abilities of language models (LMs) in reasoning and planning tasks associated with physical environments. The authors argue that LMs trained solely on large-scale text corpora lack the embodied knowledge necessary for robust performance in such tasks. To address this, they propose leveraging world models, specifically the VirtualHome simulator, to collect diverse embodied experiences and use them to construct fine-tuning tasks, such as plan generation, activity recognition, counting, and object path tracking. The proposed fine-tuning method incorporates Elastic Weight Consolidation (EWC) regularization with low-rank adaptation (LoRA) to preserve the models' generality and avoid catastrophic forgetting and make the training process more efficient.

**Strengths:**

- The paper introduces an approach to enhance off-the-shelf LLMs with embodied knowledge. The experiments involve both goal-oriented planning and random exploration. in collecting embodied experiences. While EWC and LoRA are existing methods (hence the paper may be perceived as with limited technical novelty), this work shows that their incorporation into the fine-tuning process helps retain the models' general knowledge and capabilities while adapting them to new tasks.
- The related work seems sufficient, albeit a section on catastrophic forgetting, e.g., [1] would be a plus.
- The paper provides an evaluation of the finetuned LMs on both seen and unseen tasks, and several ablation studies. The generalizability of the models is also tested, as well as their performance on the original pretraining data to ensure core language modeling abilities are retained.

[1] Korbak, Tomasz, Hady Elsahar, German Kruszewski, and Marc Dymetman. "Controlling conditional language models without catastrophic forgetting." In International Conference on Machine Learning, pp. 11499-11528. PMLR, 2022.

**Weaknesses:**

- Lack of comparison: Only two GPT-based LLMs are used. Would be valuable to consider a more diverse set of recent LLMs. The paper also does not compare the proposed E2WM paradigm with existing methods for enhancing LMs with embodied knowledge, such as [1].
- Scalability to larger models: Although the EWC-LoRA approach is designed to improve efficiency, it is not clear how well it scales to larger LM architectures. Further investigation into the scalability of the proposed method would be beneficial.
- Limitations on benchmarks: Evaluation is performed on bAbI but there are several embodied benchmarks and simulators that could be considered. Doing so would allow evaluation in realistic embodied settings, and compare other embodied models proposed (such as the Episodic Transformer [2] or [1]).
- Experimental Results: The number of tasks and models evaluated is relatively small and the presentation of the results could be clearer, e.g., adding numbers to bars in Figs. 3 and4. It seems that ChatGPT outperforms the fine-tuned models, so overall the model capacity appears to be an important factor, and the proposed fine-tuning shows marginal improvements in some of the tasks.

[1] Lin, B.Y., Huang, C., Liu, Q., Gu, W., Sommerer, S. and Ren, X., 2023, June. On grounded planning for embodied tasks with language models. In Proceedings of the AAAI Conference on Artificial Intelligence (Vol. 37, No. 11, pp. 13192-13200).
[2] Pashevich, Alexander, Cordelia Schmid, and Chen Sun. "Episodic transformer for vision-and-language navigation." In Proceedings of the IEEE/CVF International Conference on Computer Vision, pp. 15942-15952. 2021.

**Questions:**

Have the authors considered an evaluation on actual embodied benchmarks?
What is the motivation for choosing Babi as the benchmark dataset?


**Limitations:**

There is a very brief discussion on limitations as part of the conclusion section.

---

> ### Author Rebuttal · Authors · 2023-08-10
>
> Thanks for the helpful suggestions. The responses to your comments are shown below.
>
> **(1) More diverse LMs; Scalability to larger LMs**
>
> Thanks for the suggestion. We apply our approach on two larger LMs: OPT-13B and LLaMA-13B, respectively. The results on the 11 tasks (as in Figure.3 in the paper) are shown below. We can see improvements consistent to the case on smaller LMs (Figure.3). That is, our method substantially outperforms the respective base LMs while retaining a low perplexity, demonstrating the effectiveness of our method when scaling up to larger models.
>
> In addition, we’d like to note that our approach’s strong performance on **small** LMs (e.g., GPT-Neo is able to compete or even outperform ChatGPT) itself is of practical significance. And the smaller the LMs, the more significant it becomes for practical cost-efficient applications.
>
> |       Model      | Act Infer | Act Recog |   Count   | HouseQA   | NegQA     | ObjMoveQA | ObjMove   | PlanGen   | PlanGen Conf | PlanGen Unseen | PlanGen Conf Unseen | PPL     |
> |----------------|---------|---------|---------|-----------|-----------|-----------|-----------|-----------|--------------|----------------|---------------------|---------|
> |      OPT-13B     |   67.94   |   89.07   |   20.10   | 81.61     | **43.21** | **37.00** | 33.49     | 36.00     | 31.92        | 29.34          | 36.98               | 4.0768* |
> |  Ours (OPT-13B)  | **70.61** | **91.44** | **62.37** | **84.29** | 40.21     | 33.00     | **96.28** | **50.15** | **49.87**    | **45.11**      | **47.93**           | 4.3584  |
> | LLaMA-13B        | **74.05** | 90.53     | 29.38     | 81.99     | **43.21** | 28.50     | 38.82     | 41.77     | 40.33        | 38.78          | 41.73               | 3.0359* |
> | Ours (LLaMA-13B) | 68.32     | **91.80** | **79.38** | **86.59** | 30.25     | **79.00** | **96.99** | **52.05** | **51.00**    | **47.44**      | **50.49**           | 3.0690  |
>
>
> **(2) Comparison with prior LM work for embodied tasks (e.g., Lin et al.)**
>
> We want to point out the method in Lin et al. has a different goal with our work at all and thus is not comparable to our method. They aim to utilize LMs to enhance the performance on specific tasks in specific environments, thus they finetune LM to read the symbolic state and generate executable plans. However, after finetuning the model becomes a task-specialized model and loses its generality to solve various seen or unseen tasks. On the contrary, our goal is to enhance the LM itself, so the outcome of our method is still a general LM that can solve various tasks and can generalize newly acquired embodied knowledge to unseen tasks. Please refer to the general response for a more detailed explanation.
>
> **(3) Limitations on benchmarks**
>
> As we mentioned above, Lin et al. and Pashevich et al. are not comparable to our work. Evaluation in embodied environments not only evaluates the general embodied knowledge, but also tests how well the knowledge is utilized by special modules on top of the language model and how well the model is adapted to the specific environments. A strong LLM that possesses rich embodied knowledge can still fail in a specific environment since the generated action is not executable in the environment (e.g., LLM generates “Next, you should go to the kitchen” while the executable action is “<Walk> [kitchen]”)  Therefore, previous works seldom evaluate an off-the-shelf LLM in an embodied environment. They either finetune it to get a task-specialized model (e.g., Lin et al. and Pashevich et al.) or build special modules on top of the LLM [1][2]. On the contrary, we aim to evaluate the LM as a general-purpose model that can generalize new acquired knowledge to common general tasks (like QA). Please refer to the general response for more detailed reply.
>
> **(4) Experimental Numbers**
>
> To make the figure clearer, we put all the numbers in Table 6. Please refer to the appendix for them.
>
> **(4) Comparison with ChatGPT**
>
> We think our results of enhancing LMs as small as GPT-Neo (1.3B) and GPT-J (6B) to compete or even outperform the latest large ChatGPT model is surprising and important, and of practical significance. The smaller the LMs, the more significant it becomes for practical cost-efficient applications. Besides, the experiments in (1) demonstrate that our method can be scaled up to larger LMs with better performance, which further highlights the model-agnostic advantage of our method.
>
> [1] Mu et al. EmbodiedGPT: Vision-Language Pre-Training via Embodied Chain of Thought. 2023.
>
> [2] Wang et al. Voyager: An Open-Ended Embodied Agent with Large Language Models. 2023.

---

> > ### Comment · Reviewer_7bDY · 2023-08-18
> > **Thank you for your response**
> >
> > Thank you for the rebuttal and the new experiments that have addressed most of my concerns. I have raised my score to reflect this, albeit I am also not convinced that this work can be considered "embodied" in the full sense. It is worth noting that most of this work's evaluation criteria are designed to measure the performance of LMs in scenarios that do not rely on embodied context-specific physical actions or navigation within virtual worlds. General tasks such as question-answering and dialogue do not seem sufficient to claim embodiment, and while this work is valuable in broadening the general cognitive capabilities of LMs, it may be best to make it clearer that it differentiates itself from traditional embodied AI tasks, where agents operate and interact within specific environments to achieve tasks.

---

> > > ### Author Response · Authors · 2023-08-18
> > >
> > > We sincerely appreciate your valuable reviews and are glad to know that our rebuttal and new experiments have addressed most of your concerns. We agree that our evaluation criteria are different from traditional embodied AI studies. This is due to our unique goal of improving the fundamental embodied knowledge of LMs for general language problems, which has not been studied before. We will definitely make the point clearer in the revised version as you suggested!

---

### Official Review · Reviewer_ekFP · 2023-07-08

**Soundness:** 2 fair
**Presentation:** 2 fair
**Contribution:** 2 fair
**Rating:** 3
**Confidence:** 4

**Summary:**

This paper proposes to incorporate world models into large language models to enable understanding of object permanence and planning capabilities that are missing in text-only models. Specifically, the authors utilize embodied environments (VirtualHome) to collect training examples including goal-oriented planning (plan generation, activity recognition) and random exploration (counting and object plan tracking). To avoid overfitting, this paper finetunes GPT-Neo-1 3B and GPT-J-6B models using LoRA and EWC. Evaluated on seen and unseen tasks, results show that the proposed method after finetuning outperforms the corresponding text-only baselines, and match or outperform ChatGPT performance under few-shot setting.

**Strengths:**

1. This paper is motivated by an interesting limitation in text-only language models, which is the lack of embodied knowledge in pre-training. The proposed fine-tuning dataset, along with EWC-LoRA method, improve the model performance on challenging embodied reasoning and understanding tasks, without dramatically increasing the original language model perplexity.
2. The paper includes some interesting ablation studies to discuss how much improvement is observed from each fine-tuning task.

**Weaknesses:**

1. The contribution and findings of the paper is limited. Although the limitation of the LLMs is well motivated in the paper, fine-tuning on similar distribution to improve model performance and thus outperform baselines is expected. Furthermore, fine-tuning with LoRA to improve efficiency, as well as EWC to reduce overfitting have been well studied. Given that the (relatively weak) baseline is the original text-only model (and a few-shot prompted ChatGPT model), it is not convincing that the proposed method (either finetuning data or method) is generalizable to broader embodied tasks (e.g., compared to an embodied or multi-modal model).
2. Some important details are missing and the evaluation is not very convincing. The paper is phrased to incorporate world model into LLMs, but it is confusing (without explicit explanation) that both training and evaluation are not with the embodied environment. Rather, only partial of the data is collected using VirtualHome. More importantly, it is not clear how the data are constructed (for example, how to sample "irrelevant actions" for counting QA and statistics of the training and eval sets) and what the held out set is for training the model. For evaluation, I understand the automatic eval is the most convenient metic, but have been widely adapted, metrics such as Rough-L for tasks like plan generation may not reflect model performance on complex text generation tasks. See more details in the questions below.

**Questions:**

1. Have you done other evaluation on the base language model (i.e., NLP benchmarks such as superglue) apart from perplexity?
2. Why do you think EWC only outperform LoRA (in table 2)? As a regularization method, EWC constraint parameter updates and thus should understand LoRA by intuition.
3. Do you have a naive fine-tuning baseline to compare to?
4. Do you have detailed analysis on when and why the model improves performance on both seen and unseen tasks?

**Limitations:**

The paper briefly mentioned limitations.

---

> ### Author Rebuttal · Authors · 2023-08-10
>
> **(1) Limited Contribution of Finetuning**
>
> We clarify that our key technical contribution is not “finetuning”, but rather identifying and formulating the limitations of current LMs (lacking embodied knowledge), developing novel ways to automatically collect embodied experiences of desired distributions, and designing diverse finetuning tasks. To best of knowledge, no previous work has done similar studies.
>
> **(2) Finetuning on similar distribution**
>
> We clarify that we finetune and evaluate not only on the similar distribution. To test the generality of our model, we develop and collect various *unseen out-of-distribution* tasks (e.g., time reasoning in bAbI and HouseworkQA). Moreover, we believe the results that our small models (GPT-Neo/-J) compete and even outperform the much larger ChatGPT are indeed surprising, providing the first evidence of using world models to improve LMs.
>
> **(3) Novelty of LoRA and EWC**
>
> Although the specific finetuning method is not the focus of this work, we want to emphasize that our EWC-LoRA method does provide new insights.
> * **We’re the first to show that EWC for LLM finetuning is better than the previous more popular method of KL regularizor [4-6]**.
> * The recent EWC-based LLM finetuning work has only focused on in-domain training tasks. That is, they show EWC could help LLMs to remember previously trained tasks. In contrast, **we additionally show that EWC even helps with out-of-domain unseen tasks**. That is, we show the LMs finetuned with EWC obtains better performance on unseen tasks compared to finetuning without EWC (Fig.3 and Fig.4). This is because EWC effectively preverses the LM generality.
> * For LoRA, previous works usually use LoRA to improve finetuning efficiency, but **we are the first to show that combining LoRA with EWC can further prevent overfitting and improve generality** (Table 2).
>
> **(4) Comparison with embodied or multimodal model**
>
> We want to clarify that our approach has a different goal with those models, thus is not comparable to them. Please refer to the general response.
>
>
> **(5) Incorporating world model into LLMs**
>
> As above, we are not “incorporating world models into language models”, but *using world models to improve language models*. Therefore, what we finally get is still a general-purpose language model. In the training, all the used embodied experiences are collected from VirtualHome. In the evaluation, we evaluate the model on common general tasks (e.g., QA) to see if it acquires new knowledge and can generalize it to unseen tasks.
>
> **(6) Details of Dataset Construction**
>
> We have already provided the statistics of evaluation set in Section 4.1. For the training, the size of Plan Generation/Activity Recognition/Counting/Object Path Tracking is 1659/1659/1000/1000, respectively, and the held-out validation set is a Plan Generation subset of size 200. We will include all details in the revised version. For the counting QA task, as we demonstrate in line 151, all the actions (including irrelevant actions) are sampled randomly from the action space of VirtualHome.
>
> **(7) RougeL not reflecting performance**
>
> We additionally conduct human evaluations on plan generation. We follow [3] to ask 3 people to annotate whether each task can be completed using a generated plan. We randomly sample 150 tasks and ask each person to annotate 50 of them. Results are below. The higher planning accuracy demonstrates the superior task planning ability of our model.
>
> |Model|Accuracy|
> |-|-|
> |GPT-J|24.0|
> |Ours (GPT-J)|**62.4**|
>
> **(8) SuperGLUE besides perplexity**
>
> We evaluate the base GPT-J-6B and our model on appropriate SuperGLUE tasks (that can be formulated as a multi-choice QA task without prompt engineering).
>
> |Model|BoolQ|CB|RTE|AX-g|AX-b|COPA|
> |-|-|-|-|-|-|-|
> |GPT-J|45.20|**41.07**|47.29|50.00|**57.50**|59.00|
> |Ours|**66.00**|**41.07**|**58.84**|**53.37**|54.00|**62.00**|
>
> Our model’s performance matches and even outperforms the baseline, showing our model retains the general language capability.
>
>
> **(9) Why EWC outperforms LoRA**
>
> If we understand correctly, your question was: EWC adds constraints to the parameter updating, then why does EWC outperform LoRA in Table 2? (Please correct us if our understanding is not correct.) LoRA freezes the base language model and only updates a small number of parameters in adapters. We hypothesize that this can also be seen as adding constraints to the parameter update, resulting in a slightly lower performance than EWC.
>
> **(10) Naive finetuning baseline**
>
> We finetune GPT-J (GPTJ-FT) and compare with our method (GPTJ-E2WM). Our method outperforms the baseline significantly.
>
> |Model|Act-Infer|Act-Recog|Count|HouseQA|NegQA|ObjMove-QA|ObjMove|PlanGen|PlanGen-Conf|PlanGen-Unseen|PlanGen-Conf-Unseen|
> |-|-|-|-|-|-|-|-|-|-|-|-|
> |GPTJ-FT|70.99|71.41|16.49|51.34|33.33|22.50|46.25|47.98|47.59|47.86|44.43|
> |GPTJ-E2WM|**74.43**|**88.52**|**67.01**|**85.44**|**39.51**|**34.50**|**98.67**|**51.23**|**48.94**|**49.58**|**45.60**|
>
> **(11) Improvements on both seen and unseen task**
>
> When using EWC, the LM learns new knowledge from finetuning while preserving its generality. Thus, on new unseen tasks requiring the same knowledge as in the finetuning tasks, the model can utilize acquired knowledge. We’re happy to do more analysis if you have more specific questions.
>
> [1] Mu et al. EmbodiedGPT: Vision-Language Pre-Training via Embodied Chain of Thought. 2023.
>
> [2] Wang et al. Voyager: An Open-Ended Embodied Agent with Large Language Models. 2023.
>
> [3] Huang et al. Language Models as Zero-Shot Planners: Extracting Actionable Knowledge for Embodied Agents. ICML 2023.
>
> [4] Lu et al. Quark: Controllable Text Generation with Reinforced Unlearning. NeurIPS 2022.
>
> [5] Ouyang et al. Training language models to follow instructions with human feedback. 2022.
>
> [6] Liu et al. Rainier: Reinforced Knowledge Introspector for Commonsense Question Answering. EMNLP 2022.

---

> > ### Comment · Reviewer_ekFP · 2023-08-19
> >
> > Thanks for the response!
> >
> > I understand the main contribution of this paper which is to inject world knowledge into language models. I was pointing out that as one of the two main contributions highlighted in the paper (fine-tuning), it was not clear why they would perform better than fine-tuning baselines (as updated above, thank you for the results). Intuitively, and as been widely shown in previous continual learning studies, EWC and other methods reduces catastrophic forgetting, while lack behind fully fine-tuning results. However, it seems that results between GPTJ-FT and FPTJ-E2WM are the opposite. Do you have any intuition on why using E2WM performs better on the in-domain tasks?
> >
> > In terms of embodied and multimodal models, I was not looking for a direct comparison. I was mainly suggesting that as this paper claims the benefits in embodied environment (as also pointed out by other reviews), the evaluation does not seem to be convincing enough. It would be great if the author can provide some comparison to research in embodied agents in the paper revise.

---

> > > ### Author Response · Authors · 2023-08-20
> > >
> > > Thank you for your feedback. We would like to reply to your question and comment as follow:
> > >
> > > **(1) Why EWC outperforms direct finetuning**
> > >
> > > We want to clarify that our evaluation tasks are **out-of-domain** tasks instead of in-domain ones. Since our goal is to train a general-purpose model, we intentionally designed evaluation tasks that are different from the training tasks (but require similar knowledge) to test its generality. For example, as mentioned in Lines 233-234, plan generation evaluation requires *free-from natural language plans*, while the ground truth for training is *executable plans following the schema* of the specific environments. Similarly, other evaluation tasks such as Housework QA, Object Path Tracking, etc., are very different from the training tasks in nature. In addition, we also introduce diverse evaluation settings (e.g., Vanilla Unseen, Confusing Seen, and Confusing Unseen, as in Figure 3) that differs greatly from the training.  As mentioned in the paper (Lines 108, 175-182 and 194-196) and our response, EWC helps preserve the generality of LM capabilities and thus improve the generalization to out-of-domain tasks.
> > >
> > > On the contrary, previous continual learning studies with EWC typically evaluate finetuned models on in-domain tasks seen during training.
> > >
> > > **(2) Comparison to embodied agents**
> > >
> > > As we’ve clarified in the rebuttal and general response, prior works on embodied agents are not comparable to our work.  Contrary to other research on embodied agents, we've taken a unique goal and approach. Most of these studies either fine-tune the LLM to create a task-specific model [1][2] or incorporate special modules on top of the LLM [3][4], tailoring it to particular tasks. This specialization contrasts with our objective: we are striving to develop an LLM that serves as a general-purpose model enriched with embodied experiences. We’ve also demonstrated that our models can generalize new acquired knowledge to common general tasks like QA. Notably, other reviewers have acknowledged our evaluation settings (7bDY) and contributions (Reviewers cenq, DfLY and 311c). We will make the point clearer in the revised version!
> > >
> > > We sincerely thank you for your efforts in reviewing our paper and your constructive suggestions again. We hope we have resolved all the concerns, and we will deeply appreciate it if you could reconsider the score accordingly. We are always willing to address any of your further concerns.
> > >
> > > [1] Lin et al. On grounded planning for embodied tasks with language models. AAAI 2023.
> > >
> > > [2] Pashevich et al. Episodic transformer for vision-and-language navigation. ICCV 2021.
> > >
> > > [3] Shinn et al. Reflexion: an autonomous agent with dynamic memory and self-reflection. 2023.
> > >
> > > [4] Yao et al. React: Synergizing reasoning and acting in language models. 2022.

---

### Official Review · Reviewer_DfLY · 2023-07-10

**Soundness:** 4 excellent
**Presentation:** 4 excellent
**Contribution:** 3 good
**Rating:** 6
**Confidence:** 3

**Summary:**

This paper proposes a new method to improve the embodied planning capability of large language models (LLM) via adding both goal-oriented planning and random exploration data from a world model/simulator as well as the EWC-LORA regularizer that prevents catastrophic forgetting of pre-training tasks. The new regularizer EWC-LORA proposed by the authors is not only time and memory-efficient compared prior KL constraint-based regularization and also leads to negligible perplexity increase and better downstream task performance. The authors show superior performance of the method over base LLMs in constructed embodied planning and activity description tasks based on the VirtualHome simulator and also the bAbI a dataset for testing multiple types of knowledge and abilities including embodied knowledge, logic reasoning, linguistic knowledge, etc. Note that the method can only outperform much bigger LLMs such as ChatGPT in many of the scenarios.

**Strengths:**

1. The authors presents a simple yet effective way to improve LLM's embodied planning capability without sacrificing the abilities during pre-training under a small/negligible compute/memory cost. Such idea is neat and of great practical importance. I think the method will be quite significant for the future directions of making LLMs better at embodied tasks.

2. The authors have done extensive experiments in both customized and existing benchmarks/datasets to show that the proposed method can outperform base models without fine-tuning with embodied data. It is impressive that the authors show that the small models with embodied planning data fine-tuning can outperform large LLMs such as ChatGPT in many scenarios.

3. The authors also performed detailed ablation studies to show the importance of the proposed EWC-LORA regularizer via comparing to KL divergence and EWC only. The ablation studies on including various data mixture also make the paper more complete.

**Weaknesses:**

1. I think the authors should compare to stronger baselines that also consider improving language model's embodied decision-making and reasoning capabilities such as [1,2]. Improvement over base models is good but not that surprising and convincing.

2. The authors should consider other embodied decision-making benchmarks such as ALFWorld, which is used in [1. 2]. This would provide a clearer picture of the comparison between the method and prior approaches. It would also add more tasks in the empirical evaluation, which can further validate the method.

[1] Shinn, Noah, Beck Labash, and Ashwin Gopinath. "Reflexion: an autonomous agent with dynamic memory and self-reflection." arXiv preprint arXiv:2303.11366 (2023).

[2] Yao, Shunyu, Jeffrey Zhao, Dian Yu, Nan Du, Izhak Shafran, Karthik Narasimhan, and Yuan Cao. "React: Synergizing reasoning and acting in language models." arXiv preprint arXiv:2210.03629 (2022).

**Questions:**

Please address the comments listed in the section above.

**Limitations:**

Yes.

---

> ### Author Rebuttal · Authors · 2023-08-10
>
> We appreciate your positive feedback and suggestions! We would like to address your concerns in the following paragraphs:
>
> **(1) Comparison with Reflexion and React**
>
> We want to clarify that our approach has a different goal and is solving a different problem compared with the mentioned work, and our approach can be combined with these methods. Specifically, methods like Reflexion and React aim to utilize the existing embodied knowledge of LMs (which might be insufficient since they have no embodied experiences) to improve their performance on specific tasks such as navigation in MineCraft, thus they combine off-the-shelf LMs with different components like memory bank, prompting engineering, environment feedback collection, etc. On the contrary, our goal is to enhance **LM itself** by acquiring new embodied knowledge. Our final outcome is a language model with richer embodied knowledge which can still be integrated with existing methods like Reflexion and React. We aim to explore the combination in the future.
>
> In addition, we think our results of enhancing LMs as small as GPT-Neo (1.3 B) and GPT-J (6B) to compete or even outperform latest large chatGPT are significant and surprising.
>
>
> **(2) Other embodied decision-making benchmarks**
>
> The goal of our work is to inject fundamental embodied knowledge into LMs, which are not specific to particular embodied environments but are general and needed in common problems like QA and dialogue. Therefore, our evaluation is designed to assess the knowledge in those general settings (such as QA). Please refer to the general response for more details.

---

### Official Review · Reviewer_cenq · 2023-07-11

**Soundness:** 4 excellent
**Presentation:** 3 good
**Contribution:** 3 good
**Rating:** 6
**Confidence:** 4

**Summary:**

The paper proposes to enhance language models by finetuning them on “embodied experiences”, which are textual data generated by household activity simulator, which is referred to as “world models” in the paper. Through evaluations on several tasks (e.g., planning, object tracking), it is shown that finetuning on these “embodied experiences” leads to better performance compared to larger models (e.g., ChatGPT) that are not trained on these data. To minimize catastrophic forgetting and improve training efficiency, the paper also proposes to combine “elastic weight consolidation” and “low-rank adapters” to finetune the language models.

**Strengths:**

- The premise of the paper is interesting, novel, and promising — because language models are not trained on embodied data, they might be less robust to those scenarios concerning interactions with the environments (i.e., the “embodied settings”).
- The idea of using household activity simulator as a “world model” is interesting and likely significant in the context of enhancing language models.
- Moreover, the authors conduct thorough experiments that support the central claim.
- The writing and presentation of the paper are also clear.

**Weaknesses:**

Despite the strengths of the paper, below are some concerns for the problem settings and evaluations:

- While it is shown on GPT-Neo and GPT-J (which are relatively small language models in today’s standard) that the proposed approach improves their capabilities on results such as bAbI, as also indicated in the paper evaluations, larger models (i.e. ChatGPT) which are not trained on these embodied data also attain similar performance. Because the constructed tasks only require scene context in the text form, it is unclear whether similar delta will be seen on larger models. Or put this in other words: will this improvement diminish with larger scale training even without embodied experiences? The reviewer would like to note that due to practical reasons, it is understandable that experiments like these may not be done for larger LMs, but it is worth further discussions or experiments in the paper to support the claim.
- Another weakness is that the training tasks need to be hand-curated for the “embodied experience”, which includes planning, activity recognition, counting, and object path tracking. This is unlike how autoregressive LMs are trained, which only requires one main self-supervised task of next token prediction. This brings to the question of how scalable the proposed approach is, as the broader “embodied experiences” include tasks of much higher diversity, in addition to including other modalities such as vision.

**Questions:**

An important baseline is ChatGPT, which is not trained on “embodied experiences”, but it doesn’t seem like the prompt for querying ChatGPT is provided. In contrast, the paper notes that they used few-shot prompting for the smaller LMs. If they are given the same prompt, would the performance differ?

**Limitations:**

The limitations are not adequately discussed. See the comments in the “weakness” section above.

---

> ### Author Rebuttal · Authors · 2023-08-10
>
> We thank the reviewer for their positive feedback and helpful suggestions. We would like to address your concerns as follows:
>
> **(1) Scalability to larger models**
>
> Thanks for the suggestion. We apply our approach on two larger LMs: OPT-13B and LLaMA-13B, respectively. The results on the 11 tasks (as in Figure.3 in the paper) are shown below. We can see improvements consistent to the case on smaller LMs (Figure.3). That is, our method substantially outperforms the respective base LMs while retaining a low perplexity, demonstrating the effectiveness of our method when scaling up to larger models.
>
> In addition, we’d like to note that our approach’s strong performance on **small** LMs (e.g., GPT-Neo is able to compete or even outperform ChatGPT) itself is of practical significance. And the smaller the LMs, the more significant it becomes for practical *cost-efficient* applications.
>
> |       Model      | Act Infer | Act Recog |   Count   | HouseQA   | NegQA     | ObjMoveQA | ObjMove   | PlanGen   | PlanGen Conf | PlanGen Unseen | PlanGen Conf Unseen | PPL     |
> |----------------|---------|---------|---------|-----------|-----------|-----------|-----------|-----------|--------------|----------------|---------------------|---------|
> |      OPT-13B     |   67.94   |   89.07   |   20.10   | 81.61     | **43.21** | **37.00** | 33.49     | 36.00     | 31.92        | 29.34          | 36.98               | 4.0768* |
> |  Ours (OPT-13B)  | **70.61** | **91.44** | **62.37** | **84.29** | 40.21     | 33.00     | **96.28** | **50.15** | **49.87**    | **45.11**      | **47.93**           | 4.3584  |
> | LLaMA-13B        | **74.05** | 90.53     | 29.38     | 81.99     | **43.21** | 28.50     | 38.82     | 41.77     | 40.33        | 38.78          | 41.73               | 3.0359* |
> | Ours (LLaMA-13B) | 68.32     | **91.80** | **79.38** | **86.59** | 30.25     | **79.00** | **96.99** | **52.05** | **51.00**    | **47.44**      | **50.49**           | 3.0690  |
>
> **(2) Hand-Curated Training Tasks**
>
> As discussed in Introduction and Method, the diverse embodied skills are centered around two core abilities, i.e., planning and object tracking (e.g., Lines.46-50, 127-129). We collect embodied experiences based on the two abilities, and design diverse training tasks to comprehensively digest the embodied experiences. This is indeed akin to some latest LM pretraining work. For example, next-word-prediction (or sequential denoising) can be seen as training LMs to acquire the core ability of denoising; UL2 [1] shows that diversifying the training tasks (e.g., sequential denoising, span denoising, etc.) for the same core ability of denoising can substantially improve the performance, compared to using only one task (e.g., next-word-prediction). Our design of diverse training tasks follows a similar idea. To further scale up, we speculate that there could be a small set of core abilities (like planning and tracking) that facilitate the collection and design of training tasks. We’re excited to study more in the future.
>
> **(3) ChatGPT prompts**
>
> The prompts we used for ChatGPT also include in-context few-shot exemplars, as well as instructions describing the task. We will include them in the revised version.
>
>
> [1] Tay et al. UL2: Unifying Language Learning Paradigms. 2023.

---

> > ### Comment · Reviewer_cenq · 2023-08-11
> >
> > Thank you for the response and the effort for the additional experiments. I have raised the "soundness" score to 4, but I would like to maintain my overall rating given the scope considered in this work, i.e., I would still recommend acceptance for this work.
> >
> > [2] While I appreciate the effort for the extra clarification, I'm not convinced by the argument that the designed training tasks, planning and object tracking (at least in a simplified environment like VirtualHome), are sufficient to cover "embodied experiences". For example, physical  or visual experiences such as those experienced by a physical robot are also core to "embodiment" (and likely would lead to better world understanding for LLMs), but these are simply not present in the setup explored in this work. The example brought up by the authors, regarding the objectives in LLM training, is also different from the setup in this work, as those objectives are unsupervised at its core, while the training data used here are more like labeled data. However, I still like the premise proposed in this paper, and this can serve as a stepping stone for future work.

---

> > > ### Author Response · Authors · 2023-08-12
> > >
> > > Thank you so much for your supportive review!
> > >
> > > We fully agree with your points and didn’t claim planning and tracking in our work have covered all relevant “embodied experiences”. We agree that there are more diverse types of experiences (like physical and visual ones as suggested). They present enormous new opportunities for incorporation and further improvement of LMs. We hope the idea and approach presented in this work can inspire more studies in this exciting direction.

---

### Author Rebuttal · Authors · 2023-08-10

We thank all the reviewers for their insightful and encouraging comments. We are encouraged by the reviewer’s appreciation that the motivation and idea of the paper are novel, interesting and promising (Reviewers cenq, DfLY, ekFP); the proposed method is neat, effective, and of great practical importance  (cenq,DfLY, 311c), and the experiments and ablation studies are thorough and detailed (cenq, DfLY, 311c), showing strong improvements (cenq, DfLY, ekFP, 331c).

We’d like to highlight the unique focuses of our work that differ from the previous work of  embodied LMs:
* **Our goal** is to inject *fundamental* embodied knowledge and skills into LMs, such as object permanence, action planning, spatial knowledge, etc. Those fundamental embodied knowledge and skills are **not** specific to particular embodied environments (e.g., VirtualHome, ALFWorld). Instead, they are general and needed in common problems such as various question answering (QA) and dialogue. Accordingly, **our approach** aims to finetune LMs while keeping them as general-purpose models, capable of handling the common general problems (e.g., QA) and generalizing acquired knowledge to unseen tasks. Similarly, **our evaluation** assesses the knowledge/skills in those common general settings, including the novel 11 constructed tasks (mostly in QA forms) and the well-known bAbI tasks designed for assessing the fundamental knowledge/skills of models. Those evaluations are independent of specific embodied environments.
* In contrast, **prior work** on LMs for embodied tasks aims to apply the LMs to handle **specific** embodied environments, through either finetuning or prompting. Such work includes (mentioned by the reviewers) [1, 2] that finetune LMs to specific embodied environments, [3,4] that keep LMs frozen but design specialized prompts for respective environments, as well as those already discussed in our Related Work. Accordingly, their evaluation is to deploy the specialized LMs to complete specific tasks in the respective embodied environments.
* In addition, in prior work, to deploy LMs to the specific embodied environments, there are additional components needed, such as mapping the LM-generated free-form text (e.g., “*Next, you should go to the kitchen*”) into the action space (“<Walk> [kitchen]”) of the specific environment. Our work does not involve those components as we focus on the fundamental knowledge/skills and general common settings such as QA.

[1] Lin et al. On grounded planning for embodied tasks with language models. AAAI 2023.

[2] Pashevich et al. Episodic transformer for vision-and-language navigation. ICCV 2021.

[3] Shinn et al. Reflexion: an autonomous agent with dynamic memory and self-reflection. 2023.

[4] Yao et al. React: Synergizing reasoning and acting in language models. 2022.

---

### Decision · Program_Chairs · 2023-09-21

**Decision:**

Accept (poster)

**Comment:**

The authors propose an approach to finetuning LLMs with embodied experience (textual data from a simulator). They do so via EWC and LoRA for efficient training and less catastrophic forgetting. They demonstrate that the approach outperforms larger, untrained models over a variety of physical tasks in VirtualHome.

The reviewers found the paper simple yet effective and show promising experimental results, as well as detailed ablations. The approach is also timely in showing how to improve LLMs for embodied tasks, an area they have performed relatively poorly. During the rebuttal they have also included a larger model (13B). Most reviewers indicated that the paper should be accepted, and though the negative reviewer, ekFP, has some reasonable points that should be addressed, I believe the core issues are well addressed in that the evaluated tasks are out of distribution.

Overall, I think the paper is interesting and timely, and I thus recommend acceptance.

The authors should consider two additions for a final version as mentioned by the reviewers, along with the changes included in the rebuttal. (1) Adding ablations over LLM sizes (e.g., LLaMA 7B, 13B, 33B, and 65B) could establish a scaling curve and provide a significant answer to such questions. (2) More experiences or demonstrations in a second, varied environment (perhaps a manipulation environment, but generally additional embodied environments) would strongly strength the work.